

# Rebuilding soil organic C stocks in degraded grassland by grazing exclusion: a linked decline in soil inorganic C

Yi Zhang[1], Yingzhong Xie[1], Hongbin Ma[1], Le Jing[1], Cory Matthew[2] and Jianping Li[1]

[1] School of Agriculture, Ningxia University, Yinchuan, Ningxia, China
[2] School of Agriculture and Environment, Massey University, Palmston North, NewZealand

## ABSTRACT

**Background.** Our study evaluated how soil organic carbon (SOC) and soil inorganic carbon (SIC) recovered over time in deep loessial soil as overgrazed grassland was fenced and restored.

**Methods.** The study was conducted in the Yunwu Mountain Nature Reserve in the Ningxia Autonomous Region of China. In it we compared soil data from three grazed grassland (G) sites, three sites that were fenced for 15 years (F15), and three sites that were fenced for 30 years (F30) as a so-called 'space for time series'.

**Results and Discussion.** We compared SOC accumulation in soil up to 200 cm below the surface in G, F15, and F30 plots. An increase in SOC correlated with a decrease in soil pH, and decreased soil bulk density. However, SOC sequestration in fenced plots was largely offset by a decrease in SIC, which was closely correlated ($r = 0.713$, $p = 0.001$) with SOC-driven soil pH decline. We observed no significant increase in soil total carbon in the F15 or F30 sites after comparing them to G.

**Conclusions.** Our data indicate that fencing causes the slow diffusion processes to intensify the soil property changes from increased litter return, and this slow diffusion process is still active 30 years after fencing at 100–200 cm soil depths in the studied deep loessial soil. These findings are likely applicable to similar sites.

## INTRODUCTION

Grassland is one of the widest distributed vegetation types, and the type of land most affected by human activities (*Deng et al., 2016*; *Wu et al., 2016*). A major component of terrestrial ecosystems, grassland accounts for approximately 25% of global total land area and 10% of global carbon stocks, playing a vital role in the global carbon cycle (*Wang et al., 2016*). Soil carbon pools are the largest carbon pools in terrestrial ecosystems with carbon stocks about 3.8 times higher than bio-carbon stocks and three times higher than atmospheric carbon stocks. Soil carbon stock changes help regulate changes in atmospheric $CO_2$ concentration (*Fan et al., 2013*; *Ping & Wang, 2018*). Even a small change in the soil organic carbon (SOC) pool can significantly alter the atmosphere's greenhouse gas content

Corresponding author
Jianping Li, lijianpingsas@163.com

(*Dou et al., 2013*; *Srinivasarao et al., 2014*). The soil carbon pool includes both SOC and soil inorganic carbon (SIC). SOC is mainly derived from animal and plant residues, and soil microorganisms and their secretions, which are active in and provide fertility to soil. SIC is the main form of soil carbon found in arid regions and includes compounds such as soil carbonates (*Jin, Tian & Wang, 2018*; *Zhang et al., 2018b*).

China's general SOC distribution has been extensively studied. It has been noted that extremely low organic matter levels in desert grassland ecosystems may be a consequence of government-imposed grazing bans in large areas of western China, introduced in the 19th century to restore land degraded by overgrazing (*Wen et al., 2013*). The C stock of Chinese grassland vegetation has been estimated at 1.0 Pg, and the total carbon stocks of Chinese grassland ecosystems at 29.1 Pg (*Lu et al., 2018*; *Tang et al., 2018*), with grasslands accounting for 21.55% of C sequestration in Chinese terrestrial ecosystems. Grazing grassland ecosystems is beneficial for humans as well as for maintaining essential ecosystem processes such as nutrient cycling. Grazing is an important form of ecological disturbance and control factor in preserving equilibrium in natural grassland ecosystems (*Piñeiro et al., 2009*). However, overgrazing can cause severe degradation of desert steppes or other grassland types, and can further reduce SOC below its already naturally-reduced levels caused by low precipitation and other environmental factors (*Deng et al., 2016*; *Wang et al., 2016*). Similarly, a reduction of SOC content in alpine meadow soils is also correlated with a degree of degradation (*Yuan et al., 2019b*). Restoring degraded grassland can increase grassland ecosystem carbon stocks, particularly soil C stocks. Higher soil C enhances the respiration of grassland soils and accelerates the decomposition of soil organic matter (*He et al., 2019*; *Zhou et al., 2019*).

Developing indices to measure degradation and guide restoration project management is an extremely complex undertaking. In a study of natural wetland restoration in Florida's Everglades, *Doren et al. (2009)* drew inspiration from existing restoration projects in California and the Chesapeake Bay to choose 11 ecological indicators for monitoring, based on 12 selection criteria. In a study of rangeland restoration in Iceland, researchers examined the importance of stakeholder attitudes towards restoration and the function of government policies on the achieved outcomes (*Petursdottir et al., 2013*). Following widespread government-implemented overgrazing of rangeland in the late 20th century aimed at increasing food supply, rangeland grazing in China has been banned since the early 2000s in many provinces to promote ecosystem recovery. However, many farmers have mixed feelings about the current grazing bans and do not always abide by them. In response, the government introduced both enforcement and educational initiatives to inform farmers about the advantages of fencing (*Fan et al., 2013*; *Ping & Wang, 2018*). In years since, a number of studies have investigated the recovery of various restoration plots after fences were installed (*Li et al., 2014*; *Li et al., 2013*; *Zhang et al., 2018a*).

Fencing allows for the recovery of both above- and below-ground rangeland ecosystem components, including vegetation diversity, resilience, and physical and chemical soil properties (*Yuan & Hou, 2014*; *Yuan et al., 2019a*). Fencing eliminates the feeding and trampling effects of livestock (unless incursions occur), and as a result vegetation and biomass typically recover quickly from rare incursions (*Li et al., 2019*). Ecosystem litter

flow then increases and enters the soil, increasing the amount of soil carbon. Additionally, the restoration of vegetation cover protects surface SOC, reducing soil erosion, increasing the interception of wind-blown dust and other fine particles in the atmosphere (*Ping & Wang, 2018*), and promoting SOC (and in some cases, SIC) retention (*Jin, Tian & Wang, 2018*; *Wu et al., 2016*). Some results indicate that SIC storage deceases in restored grasslands because of the decrease in soil pH and increase in soil water content. SIC plays an important role in carbon sequestration and vegetation restoration in the semi-arid Loess Plateau (*Liu et al., 2014*). Other studies found increased SIC among all studied land-use type changes (grassland to farmland, grassland to forest, farmland to grassland, farmland to forest, and sandland to forest), while the magnitude and direction of SIC changes varied greatly with different land-use types (*An et al., 2019a*; *An et al., 2019b*).

Rangeland provides degraded grasslands with significant carbon sequestration potential (*An et al., 2019a*; *An et al., 2019b*). that should be utilized as an ecosystem service. For example, if grassland's carbon sequestration rate was between 46.7–129.2 g cm$^{-2}$ yr$^{-1}$ with an average of 84.2 g cm$^{-2}$ yr$^{-1}$, the average net carbon stocks would increase by 130.4 g m$^{-2}$ yr$^{-1}$ (*Wang et al., 2010*). Related research conducted in China's semiarid steppe grassland of Inner Mongolia found that the plant-soil carbon stocks of fenced grassland were 1.5 times greater than those of grazed grassland (*Hoffmann et al., 2016*; *Yang et al., 2017*).

A number of recent studies have investigated SOC recovery following fencing and grazing exclusion, but none of these explored soil depths below 100 cm. In order to further examine the recovery of overgrazed grassland, we studied the recovery time course of both SOC and SIC, as well as the interactions between the two deep soil carbon pools (0–500 cm). Since few studies on C cycling and post-fencing soil C recovery have included SIC or explored changes below 100 cm soil depth, our data will provide greater clarity on the impacts of land husbandry changes on C sequestration rates, both in China and abroad.

## MATERIALS & METHODS

### Study sites

The studied site was located in the Yunwu Mountain National Nature Reserve (36°10′–36°17′N, 106°21′–106°27′E), 45 km northeast of Guyuan City, Ningxia Autonomous Region, China in the hinterland of the Loess Plateau at an altitude of 1,700–2,148 m. It belongs to the temperate semi-arid climate zone. The mean annual precipitation for the period 1,980–2,014 was 425 mm, with 60%–75% of total precipitation occurring between July and September. The main soil types are locally known as 'mountain gray cinnamon' soil and 'black loess', which are Alfisols and Mollisols, respectively, according to the USDA classification system. Atmospheric precipitation mainly replenishes water resources. The region's dominant plants are *Stipa bungeana, Stipa grandis, Thymus mongolicus, Artemisias acrorum,* and *Potentilla acaulis.*

Since 1980, the regional government has implemented several mountain closure and grazing prohibition measures, closing plots to livestock grazing and making them available
**Table 1 Location of studied sites.** The Yanzhou District Grassland Management Station, located in Guyuan City, identified sites that were grazed (G) or fenced for 15 (F15) or 30 (F30) years.

| Fenced period | Longitude | Latitude | Altitude (m) | Ground cover (%) | Slope (°) | Slope aspect |
|---|---|---|---|---|---|---|
| | 106°24′13″E | 36°10′04″N | 1,761 | 40 | 17–19 | Semi-sunny slope |
| G | 106°24′13″E | 36°10′01″N | 1,795 | 30 | 0–1 | Sunny slope |
| | 106°24′11″E | 36°10′00″N | 1,788 | 38 | 16–18 | Semi-sunny slope |
| | 106°22′53″E | 36°13′31″N | 1,910 | 85 | 6–8 | Semi-sunny slope |
| F15 | 106°23′10″E | 36°13′32″N | 1,940 | 95 | 0–1 | Sunny slope |
| | 106°23′14″E | 36°13′27″N | 1,954 | 97 | 5–7 | Semi-sunny slope |
| | 106°22′53″E | 36°15′07″N | 2,077 | 100 | 0–1 | Sunny slope |
| F30 | 106°23′10″E | 36°15′03″N | 2,048 | 100 | 13–15 | Semi-sunny slope |
| | 106°23′14″E | 36°16′02″N | 2,112 | 100 | 8–10 | Semi-sunny slope |

**Notes.**
The number of years ($\pm$ 1–2 years) that sites were fenced in was determined in 2014.

for study. We selected sites in the Yunwu Mountain Nature Reserve that had similiar soil and vegetation types, were distributed over a distance of up to 45 km with a rolling topography, and had been fenced for 15 or 30 years (F15 and F30, respectively) together with similar sites still being grazed (G). There were three replicate sites in each case. Site information appears in Table 1.

## Experiment design and sampling

We designed to investigate the differences among the G, F15, and F30. Three 10 m ×10 m plots were set for each level, with nine sites in total. At each of the nine sites, we selected a homogeneous sampling area deemed visually representative of the site with an area of approximately 900 m$^2$. Three 1×1 m square quadrats were placed at 10 m intervals along a transect line for sampling. A six cm diameter auger (S1 Canada) was used to extract soil samples within each quadrat (three samples per quadrat) to a 500 cm depth. Soil samples were packed into Ziploc bags on site for processing in the laboratory. The 0–40 cm soil depth samples were collected in 10 cm depth increments, and samples from 20 cm depth increments were collected from 40–500 cm, making a total of 27 samples per core. To find the bulk density (BD), we used a 51-mm soil sampling drill (S1 Canada) to obtain an undisturbed soil core from 0 to 500 cm, and a soil bulk sampler method to measure the BD (g cm$^{-3}$) of the different soil layers (0–10, 10–20, 20–30, 30–40 cm; from 40 to 500 cm, samples were taken every 20 cm). In the undisturbed soil cores from the FG and G plots, the sampler was 100 cm$^3$ and the sampler's inner diameter was 50 mm (*Li et al., 2019*). In the laboratory, we determined the subsamples' soil moisture content, then sample remainders were air dried and passed through a two mm sieve to remove mixed litter and roots. Nine cores per sampling site were then bulked together for each soil layer. The analyses for SOC, SIC, and pH were performed in triplicate on subsamples from the bulked site samples. SOC and SIC were measured using an elemental analyzer (Vario EL/micro cube, Germany) and soil pH value was measured using an acidity agent (PHS-3C pH acidometer, China).

## Soil carbon calculations

The SOC and SIC stocks (Mg hm$^{-2}$) were calculated as follows (*Li et al., 2019*):

$$SOC\ storage = \sum_{i=1}^{n} D_i \times B_i \times SOC_i \tag{2.1}$$

$$SIC\ storage = \sum_{i=1}^{n} D_i \times B_i \times SIC_i \tag{2.2}$$

Where n is the number of soil layers, $D_i$ is the soil depth (cm), $B_i$ is the soil bulk density (g cm$^{-3}$), $SOC_i$ and $SIC_i$ are the SOC content and SIC content (%) of soil layer$_i$.

The soil total carbon (STC) stocks were calculated as:

$$STC\ storage = SOC\ storage + SIC\ storage \tag{2.3}$$

And the changes in soil C stocks (C change, Mg hm$^{-2}$) were:

$$C\ change = C_t - C_0 \tag{2.4}$$

Where $C_t$ represents soil stocks in the fenced grassland (Mg hm$^{-2}$), and $C_0$ is the soil stocks in the grazed grassland (Mg hm$^{-2}$).

Lastly, the rate of soil carbon stock change (RSS, Mg hm$^{-2}$ yr$^{-1}$) was estimated by the linear regression of change in soil C stocks with time since fencing:

$$C\ change = f(\Delta time) = y_0 + K \times \Delta time \tag{2.5}$$

Where $y_0$ is a constant, $K$ is the rate of soil C stock change (Mg hm$^{-2}$ yr$^{-1}$), and $\Delta\ time$ is the fenced time interval (i.e., years from grazing cessation to the present).

## Statistical analyses

Various statistical analyses were performed using the software packages SAS 9.4, Origin 2017, and Microsoft Excel. Since soil depth data taken from the same cores can suggest correlation across soil depths, all data were first analyzed using One-Way ANOVA comparing the time spent fenced off with soil depths as a statement of "mixed effect model" in Proc GLM in SAS (*Li et al., 2019*). For presentation and statistical purposes, the bulked samples from the nine sites (each consisting of 27 soil layers) were averaged across each of the six soil layers (0–40, 40–100, 100–200, 200–300, 300–400, and 400–500 cm soil depths). The soil depth effect had a degrees of freedom (df) of 5; the interaction effect between time fenced and soil depth had a df of 10, with an error df of 30 and a total of 53 (9× 6–1) df in each repeated measures ANOVA. SAS provides two options for adjusting the *p*-value of the main-factor ×repeat-factor interaction across soil depths, and we used the adjusted *p*-value based on the Greenhouse-Geiser Epsilon. We then processed the SAS repeat measure output using the other packages. For example, we used SPSS to obtain the standard errors of individual means across soil depths, and Minitab Version 10.51 to perform Pearson correlation analyses.

## RESULTS

### Soil bulk density, soil moisture (SWC), and pH

The observed soil bulk density values ranged from 0.91 g cm$^{-3}$ in the 0–40 cm soil layer of the F15 plots to 1.67 g cm$^{-3}$ in the 100–200 cm layer of the F30 plots. For subsoil with low SOC, typical values were 1.2–1.28 g cm$^{-3}$. Values for the 0–40 cm layer were lower, reflecting the accumulation of SOC in the upper soil. For the grazed plots, the mean bulk density in the 0–40 cm soil depth was 1.09 g cm$^{-3}$, and was reduced to less than 1.00 g cm$^{-3}$ in the F15 and F30 plots (Table 2). The results from repeated measures ANOVA showed that time fenced and soil depth both significantly affected soil bulk density ($F = 23.10$, $p < 0.0001$; Table 3). The high bulk density of 1.67 g cm$^{-3}$ in the 100–200 cm layer of the F30 plots was atypical, since all the others were observed at less than 1.30 g cm$^{-3}$, so we noted a correlation of –0.802 ($p = 0.008$) between soil bulk density and total carbon (TC) (Table 4). The pattern of variation in soil water content (SWC) generally followed the pattern of variation in SOC, with a correlation of $r = 0.849$ (Table 4; $p < 0.001$). Thus, SWC was higher in the surface horizon than in deeper horizons, and lower in grazed than in F15 and F30 grasslands (Table 2). Although SWC was (not unexpectedly) higher in the 100–300 cm soil horizon than in the 300–500 cm soil horizon (9.4% and 10.3% on average, respectively; $p = 0.399$), SOC values were lower at the same depths (0.47% and 0.29%, respectively; $p = 0.362$).

All soil pH values were alkaline, ranging from the lowest observed value of 7.76 in the 0–40 cm soil layer of F30 plots to 9.20 and 9.18 in the 200–300 and 300–400 cm soil layer of grazed plots. Notably, across soil depth and fencing duration data, pH values were strongly negatively correlated with SOC and strongly positively correlated with SIC ($r = -0.890$, $p < 0.001$ and $r = 0.713$, $p = 0.001$, respectively; Table 4). pH was consistently lower in F15 and F30 plots than G plots in soil up to 400 cm deep.

### Soil organic and inorganic carbon

In grazed grassland, SOC g kg$^{-1}$ ranged from 14.3 g kg$^{-1}$ in the 0–40 cm soil horizon to 3.1 g kg$^{-1}$ in the 400–500 cm soil horizon ($p < 0.05$). In F30 grassland, the corresponding values were 22.9 g kg$^{-1}$ (a 60% increase compared to G, $p < 0.05$) and 3.1 g kg$^{-1}$ (unchanged). For the soil depth of 200–300 cm, F15 had lower SOC levels than grazed grasslands but F30 had the highest, and for the 300–500 cm soil depth, the SOC levels of F15 were not significantly different than grazed grasslands but F30 had higher levels (Table 5). By contrast, SIC values changed much less with soil depth, but were higher in the deep soil layers compared to surface levels in grazed sites ($p < 0.05$). The opposite was true for F30 and F15 grassland when compared to grazed grassland (grazed 0–40 cm soil depth, 17.1 g kg$^{-1}$ SIC; grazed 400–500 cm soil depth, 18.9 g kg$^{-1}$ SIC; F30 0–40 cm soil depth, 10.8 g kg$^{-1}$ SIC; F30 400–500 cm soil depth, 16.9 g kg$^{-1}$ SIC) (Table 6; $p = 0.0046$ for the GG adjusted, time fenced ×soil depth interaction). Repeated measures ANOVA showed that time fenced and soil depth both significantly affected SOC content ($F = 502.4$, $p < 0.0001$; Table 3).

The SOC and SIC data are presented as total carbon stocks (Mg hm$^{-2}$) in Figs. 1–3 show the changes relative to grazed sites. These data illustrate that the SOC accumulation at this set of sites extended below 100 cm soil depth and continued 15 and 30 years after fencing,

**Table 2  Soil bulk density (BD), soil water content (SWC), and soil pH for 0–500 cm soil depths in grazed grassland (G), restored grassland, and areas fenced off for 15 (F15) or 30 (F30) years.** Changes in BD, SWC and pH content with years fenced and soil depth ($n = 3$ sites for each mean value).

| | | | Soil depth (cm) | | | | | |
|---|---|---|---|---|---|---|---|---|
| | | | 0–40 | 40–100 | 100–200 | 200–300 | 300–400 | 400–500 |
| BD (g cm$^{-3}$) | G | Mean | 1.09aC | 1.12aC | 1.21aB | 1.23aB | 1.26aAB | 1.28aA |
| | | SEM | 0.04 | 0.04 | 0.02aB | 0.02 | 0.01 | 0.01 |
| | F15 | Mean | 0.91bD | 1.07bC | 1.23bA | 1.22aB | 1.23bA | 1.28aA |
| | | SEM | 0.01 | 0.01 | 0.01 | 0.01 | 0.01 | 0.01 |
| | F30 | Mean | 0.95bE | 0.99cD | 1.67bA | 1.23aC | 1.24bB | 1.26bB |
| | | SEM | 0.01 | 0.01 | 0.01 | 0.02 | 0.01 | 0.01 |
| SWC (%) | G | Mean | 12.34bA | 10.53cAB | 7.25bC | 8.14bBC | 9.67aBC | 10.03aB |
| | | SEM | 0.01 | 0.01 | 0.01 | 0.01 | 0.01 | 0.01 |
| | F15 | Mean | 14.91aA | 14.94aA | 8.73bB | 9.37bBC | 10.67aB | 11.81aB |
| | | SEM | 0.01 | 0.01 | 0.01 | 0.01 | 0.01 | 0.01 |
| | F 30 | Mean | 14.79aA | 13.99bA | 11.62aC | 10.99aBC | 10aC | 9.85aC |
| | | SEM | 0.01 | 0.01 | 0.01 | 0.01 | 0.01 | 0.01 |
| pH | G | Mean | 8.62aB | 8.83aB | 8.92aA | 9.2aA | 9.18aA | 8.97aA |
| | | SEM | 0.05 | 0.09 | 0.11 | 0.01 | 0.05 | 0.33 |
| | F 15 | Mean | 8.04aB | 8.71aA | 8.84aA | 8.85bA | 8.85bA | 8.59aA |
| | | SEM | 0.65 | 0.14 | 0.08 | 0.11 | 0.13 | 0.22 |
| | F30 | Mean | 7.76bB | 8.33bA | 8.63bA | 8.89bA | 8.88bA | 9.04aA |
| | | SEM | 0.18 | 0.07 | 0.1 | 0.1 | 0.1 | 0.04 |

Notes.
Different lowercase letters indicate significant ($p < 0.05$) differences between fencing times; different uppercase letters indicate significant differences ($p < 0.05$) between soil depths.

**Table 3  F and *p*-values from SAS repeated measures ANOVA in Proc GLM.** The effect time fenced (T), soil depth (D), the interaction between T and D (T*D) on soil bulk density (BD), soil gravimetric water content (SWC), soil pH, soil organic carbon (SOC) content, soil inorganic carbon (SIC) content, and soil total carbon (STC) content.

| | T | | D | | T*D | |
|---|---|---|---|---|---|---|
| | $F_{2,6}$ | $p$ | $F_{5,30}$ | $p$ | $F_{10,30}$ | $p$ |
| BD (g cm$^{-3}$) | 15.60 | 0.004 | 515.5 | <0.0001 | 23.10 | <0.0001 |
| SWC (%) | 10.47 | 0.011 | 50.21 | <0.0001 | 5.09 | 0.0002 |
| pH | 5.18 | 0.049 | 15.96 | 0.0002 | 1.00 | 0.4465 |
| SOC (%) | 4,390 | <0.0001 | 10,324 | <0.0001 | 502.4 | <0.0001 |
| SIC (%) | 18.16 | 0.0028 | 9.14 | 0.0081 | 8.43 | 0.0046 |
| STC (%) | 1.94 | 0.224 | 94.56 | <0.0001 | 2.04 | 0.1750 |

Notes.
$F_{2,6}$, $F_{5,30}$, and $F_{10,30}$ indicate numerator and denominator degrees of freedom, respectively, of the F statistics for each ANOVA effect. For D and T*D, $p$-values are based on the Greenhouse-Geisser epsilon option in SAS.

as evidenced by a F15–F30 separation and positive movement relative to grazed plots at the 0–200 cm soil depth (Figs. 2A, 3A). The rate of change for years 0–30 was different from the rate of change for years 0–15, particularly in the 0–200 cm soil depths where SOC accumulation occurs. In all three of the upper soil layers (0–40, 40–100, and 100–200 cm

**Table 4 Correlations between soil properties representing the effects of years fenced and soil depths.**

|  | SOC | SIC | TC | BD | SWC |
|---|---|---|---|---|---|
| SIC | −0.738[*] |  |  |  |  |
| TC | 0.879[**] | −0.325 |  |  |  |
| BD | −0.883[**] | 0.613[*] | −0.802[**] |  |  |
| SWC | 0.849[**] | −0.637[*] | 0.739[**] | −0.775[**] |  |
| pH | −0.890[**] | 0.713[*] | −0.742[**] | 0.816[**] | −0.812[**] |

Notes.
[*]represent significant correlation at the lever of $P < 0.05$.
[**]represent significant correlation at the lever of $P < 0.01$.

**Table 5 Soil organic carbon (SOC, g kg$^{-1}$) for 0–500 cm soil depths in grazed grassland (G) and in grassland under restoration, fenced for 15 (F15) or 30 (F30) years.** Changes in SOC content with years fenced and soil depth ($n = 3$ sites for each mean value).

| | | Soil depth (cm) | | | | | |
|---|---|---|---|---|---|---|---|
| SOC (g kg-1) | | 0–40 | 40–100 | 100–200 | 200–300 | 300–400 | 400–500 |
| G | Mean | 14.3cA | 7.2cB | 2.6cD | 3bC | 2.7bD | 3.1bC |
|   | SEM | 0.26 | 0.07 | 0.08 | 0.05 | 0.04 | 0.11 |
| F15 | Mean | 17.7bA | 12.4bB | 4.6bC | 2.7cE | 2.6 bE | 2.9cD |
|   | SEM | 0.02 | 0.13 | 0.01 | 0.08 | 0.07 | 0.02 |
| F30 | Mean | 22.9aA | 16.2aB | 11.7aC | 3.7aE | 3.2aE | 3.1aD |
|   | SEM | 0.04 | 0.01 | 0.04 | 0.26 | 0.09 | 0.03 |

Notes.
Different lowercase letters indicate significant ($p < 0.05$) differences between fencing times; different uppercase letters indicate significant differences ($p < 0.05$) between soil depths.

**Table 6 Soil inorganic carbon (SIC, g kg$^{-1}$) from 0–500 cm soil depths in grazed grassland (G) and in grassland under restoration, fenced for 15 (F15) or 30 (F30) years.** Changes in SIC content with years fenced and soil depth ($n = 3$ sites for each mean value).

| | | Soil depth (cm) | | | | | |
|---|---|---|---|---|---|---|---|
| SIC (g kg$^{-1}$) | | 0–40 | 40–100 | 100–200 | 200–300 | 300–400 | 400–500 |
| G | Mean | 17.1aB | 20.5aA | 15.3aA | 15.4aA | 17.3aA | 18.9aA |
|   | SEM | 0.08 | 0.01 | 0.06 | 0.04 | 0.21 | 0.01 |
| F15 | Mean | 9.1bB | 12.9bB | 16.1aA | 16.4aA | 16.7aA | 16.2aA |
|   | SEM | 0.02 | 0.03 | 0.11 | 0.01 | 0.06 | 0.02 |
| F30 | Mean | 10.8bB | 9.6bB | 10.2bB | 17.9aA | 16.6aA | 16.2aA |
|   | SEM | 0.03 | 0.07 | 0.05 | 0.14 | 0.18 | 0.03 |

Notes.
Different lowercase letters indicate significant ($p < 0.05$) differences between fencing times; different uppercase letters indicate significant differences ($p < 0.05$) between soil depths.

soil depths), SOC accumulation at the 40–100 cm soil depth occurred faster in the first 15 years after fencing, and occurred mainly in years 15–30 at the 100–200 cm soil depth. A reciprocal trend was seen in SIC data, but because of greater variability in this data, the trend could not be confirmed as statistically significant.

After comparing the SOC and SIC data, the effects of fencing on SIC stock were considered non-significant below 200 cm soil depths. However, in the 0–200 cm soil depth,

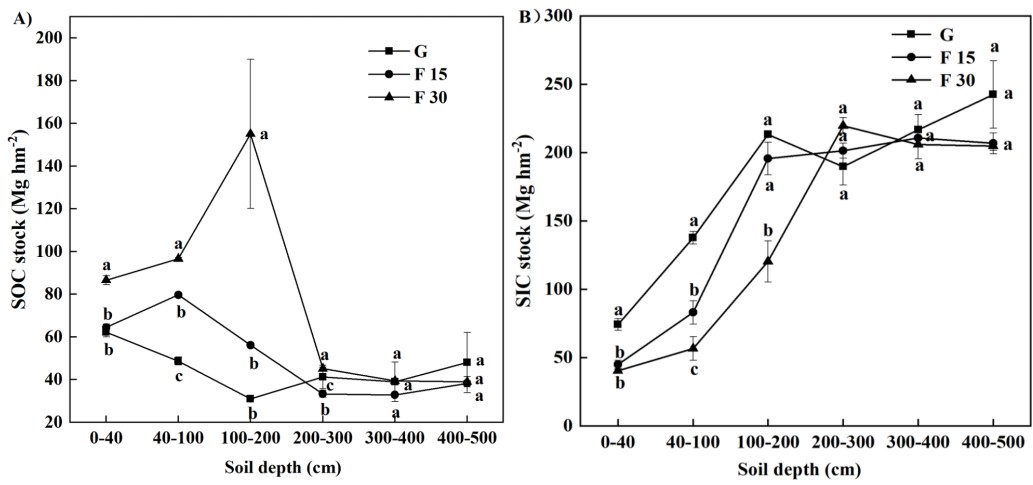

**Figure 1** **The vertical distribution of soil carbon stocks for grazed grasslands (G), and for restoring grasslands that had been fenced for 15 (F15) or 30 (F30) years, respectively.** (A) soil organic carbon (SOC), and (B) soil inorganic carbon (SIC) stocks (Mg hm$^{-2}$). Different lowercase letters indicate significant ($p < 0.05$) differences between G, F15, and F30 sites.

the SIC stock decrease was larger than the SOC stock increase, to the extent that total soil C stocks (SOC stock + SIC stock; TC) across the 0–500 cm sampled soil profile were lower at fenced sites compared to grazed sites (Grazed 1345 Mg hm$^{-2}$; F30 1310 Mg hm$^{-2}$; $p = 0.0479$).

## Total carbon stock percentages of the SOC and SIC pools

The average SOC stock was 270 Mg hm$^{-2}$ in grazed grassland, 304.7 Mg hm$^{-2}$ in F15, and 461.6 Mg hm$^{-2}$ in F30 grasslands. SOC stock significantly increased at the F30 sites, compared to the F15 sites, while the SIC stock showed an opposite trend. SIC stocks decreased with time fenced, suggesting that STC did not significantly differ with years fenced (Fig. 4).

SOC stocks for the 0–40 cm soil depth significantly increased at F30 but not F15 sites (Fig. 5A), while at 40–100 cm and 100–200 cm soil depths, SOC was significantly greater ($p < 0.05$) in F15 than in grazed sites, and significantly greater ($p < 0.05$) again in F30 than in F15 sites (Figs. 5B, 5C). In contrast, SIC stocks were lower in F15 and F30 sites compared to grazed sites at all soil depths, and the reduction was greater for F30 than F15 sites at deeper soil depths of 40–100 and 100–200 cm ($p < 0.05$; Figs. 5D–5F). When observing the combined effect of SOC and SIC changes, the STC for the 0–40 cm soil depth was reduced at F15 sites compared to grazed sites but had recovered at F30 sites; at 40–100 cm soil depths, the STC was reduced at F15 and F30 sites compared to Grazed sites; and at 100–200 cm soil depths, the STC did not differ among F15, F30, and grazed sites (Figs. 5G–5I).

## DISCUSSION

China has some of the largest natural grasslands in the world. As previously mentioned, many of these grasslands were overgrazed in the latter part of the 20th century, which has

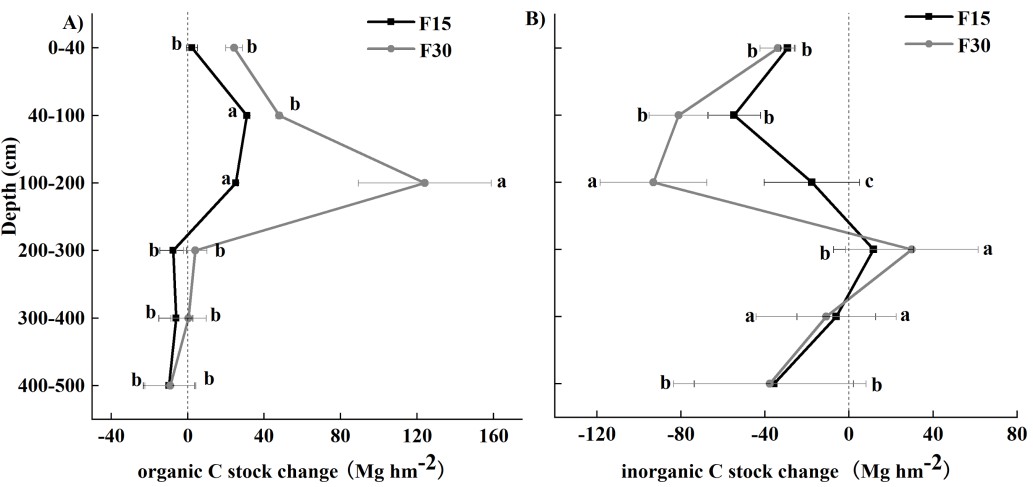

**Figure 2** Carbon stock change differences between grazed Loess Plateau grassland and grassland fenced for 15 (F15) or 30 (F30) years, respectively, for vertical distribution. (A) Soil organic carbon, (B) soil inorganic carbon. Different lowercase letters indicate significant ($p < 0.05$) differences among the different soil depths.

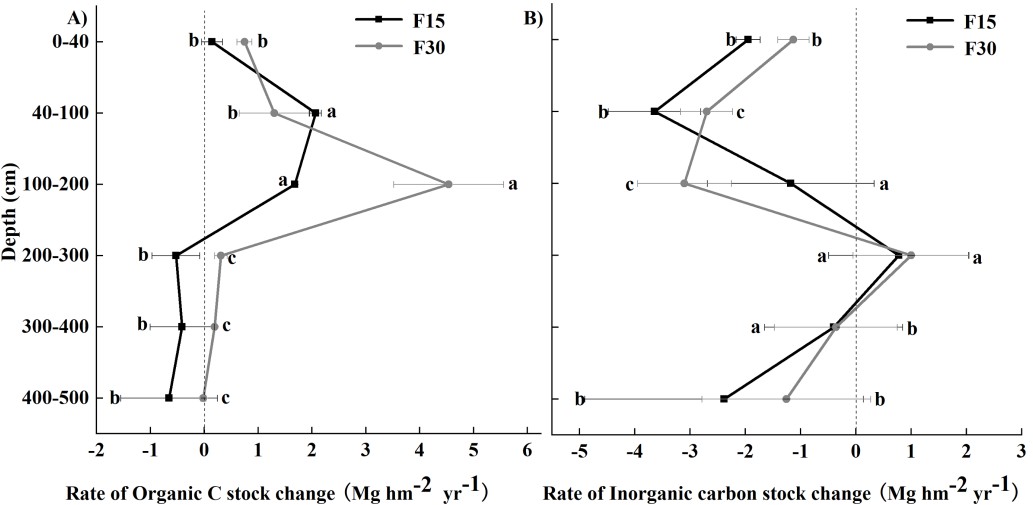

**Figure 3** Rates of carbon change relative to grazed grassland and grasslands fenced for 15 (F15) or 30 (F30) years, respectively. (A) Rate of soil organic carbon stock change and (B) Rate of soil inorganic carbon stock change. Error bars represent standard errors of means. Different lowercase letters indicate significant ($p < 0.05$) differences among the different soil depths.

encouraged the planning of regional grassland recovery programs. The soils of China's Loess Plateau, lying in the middle reaches of the Yellow River, were formed from aeolian deposits accumulated over millennia and are sometimes more than 100 m thick (*Li et al., 2019*; *Liu et al., 2020*). From a soil formation perspective, this suggests the absence of a stage involving the weathering of parent material such as rock, with the parent material evident at a certain depth, and a soil matrix transition with depth reflecting the gradation

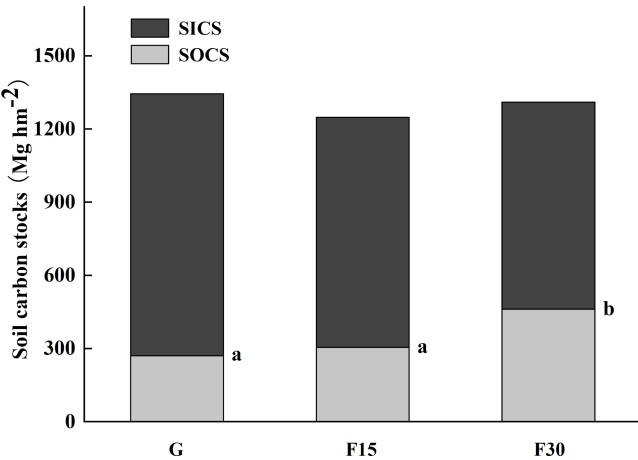

**Figure 4  Soil organic carbon (SOC) and inorganic carbon (SIC) stocks over the total 0–500 cm soil profile for grazed grassland (G) and grassland fenced for 15 (F15) or 30 (F30) years, respectively.** Different lowercase letters indicate significant ($p < 0.05$) differences in SOC with time fenced.

from soil to parent material. This raises the question: to what depth will processes like soil C regeneration penetrate during the restoration of the homogeneous loess matrix? Previous studies have explored SOC changes over time under different land uses (*Chen et al., 2017*), SOC decline under continued grazing (*Yuan & Hou, 2014*), and SOC increase after the Loess Plateau's steppe grassland was fenced off to protect from grazing (*Liu et al., 2017*; *Zhang et al., 2015*). That final study also showed that SOC increase during grassland restoration is accompanied by SIC decrease, and they concluded that both SOC and SIC should be measured when compiling soil C inventories for regional development planning (*Zhang et al., 2015*). Additionally, Chen et al. studied sites for 3 years and detected statistically significant changes in soil properties within that period (*Chen et al., 2017*). We chose sites that formed a chronosequence to observe changes over a longer time period, which is a standard method used in China (*Liu et al., 2017*; *Zhao et al., 2019*).

SOC levels at the grazed sites were low and the pH was somewhat alkaline (SOC 14.3 g kg$^{-1}$ and 7.2 g kg$^{-1}$ for 0–40 and 40–100 cm soil depths, respectively; with corresponding pH values 8.62 and 8.83), reflecting the cold, arid climate of the Loess Plateau and modest litter returns of a plant community with a net primary productivity of around 3–5 Mg hm$^{-2}$ year$^{-1}$ dry matter that is subject to grazing. However, our data confirm the beneficial changes in soil properties that follow grazing exclusion reported in other studies, with a 60% increase in SOC levels for the 0–40 cm soil depth at the F30 sites, decreased soil BD, decreased alkalinity, and increased SWC. Our sampling also extended to 500 cm depths and we observed major SOC accumulation in the 100–200 cm soil horizon (Table 5; Fig. 5), which was not detected by other studies only using a 100 cm sampling depth.

We also confirmed the negative correlation between SOC increase and SIC decrease over time (*Liu et al., 2017*) (Fig. 4). When the means (Tables 2, 5 and 6) for the six measured soil depths and years fenced (grazed, F15, or F30) were compiled into a correlation matrix (Table 4), the correlation coefficient for SOC and SIC was −0.738 ($p < 0.001$). Considering

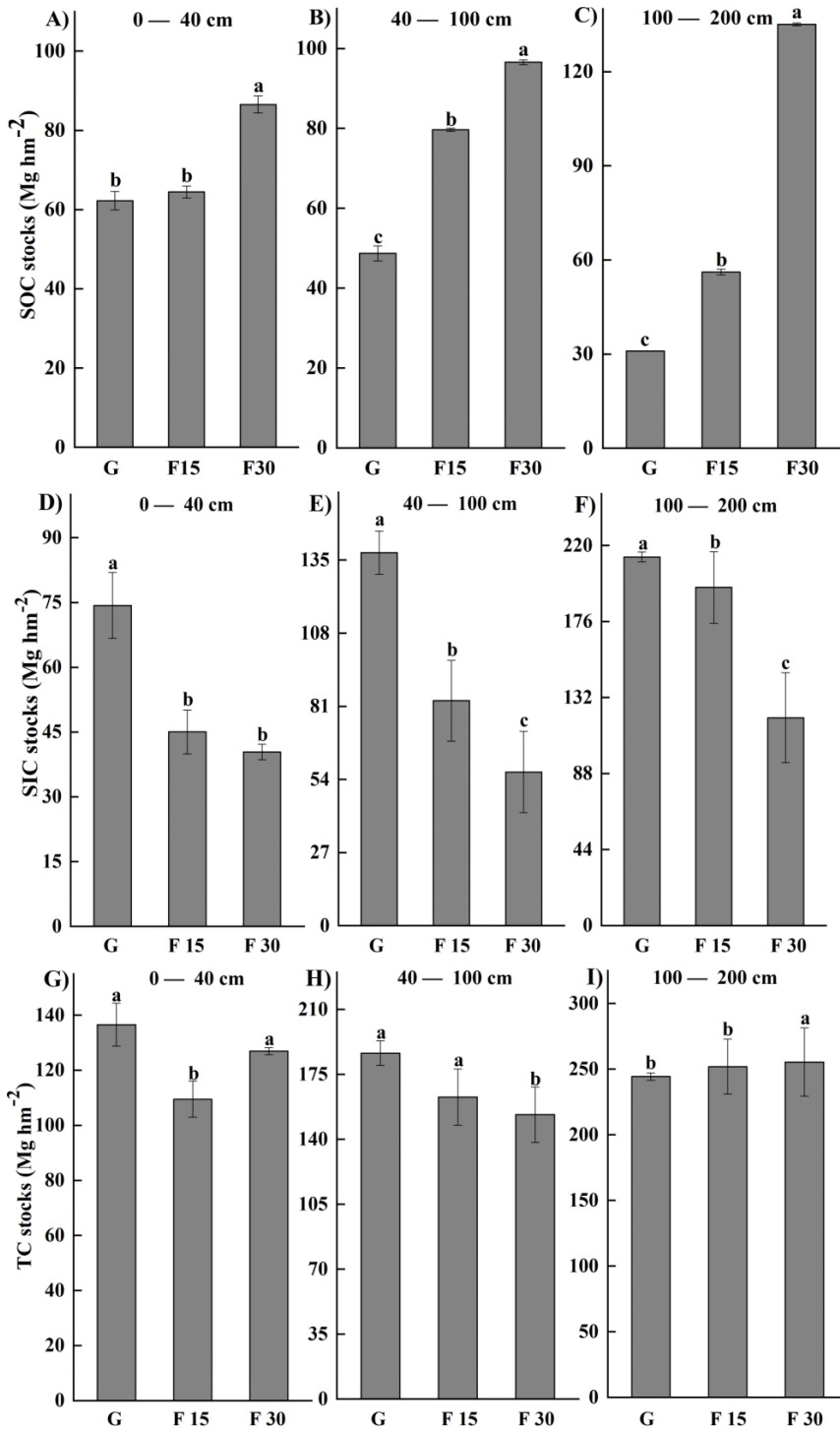

**Figure 5** **Changes in three soil depth bands (0–40, 40–100, 100–200 cm) for soil organic carbon (SOC) (A–C) □ soil inorganic carbon (SIC) (D–F), and soil total carbon (STC) stock (G–I).** In the x-axes, G, F15, and F30 denote grazed grassland and grassland fenced for 15 or 30 years, respectively. Note: upper-case letters with each error bar indicate significant (p < 0.05) changes with length of time fenced.

the linked correlations between SOC and other soil properties (Table 4), particularly the SOC–pH correlation of -0.890, we concluded that the SOC–SIC antagonism in these soils is a shift in the chemical balance shown in eq1.1 (*Liu et al., 2017*) away from carbonate formation through the SOC-induced reduction in alkalinity (i.e., pH decrease).

An important factor in restoring grassland degraded by historic overgrazing and in predicting the greenhouse gas implications of environmental management policies is the time frame for recovery. Our SOC data were extremely consistent across sites, making the SOC accumulation shown in years 0 to 15 of the site's chronosequence significantly different from years 15 to 30.

Further research will be needed to determine if the pH-mediated loss of SIC associated with SOC build up after fencing applies to the majority of soil types or if it is specific to soils with particular parent materials or properties. However, it is clear that predicting the greenhouse gas implications of land restoration programs is not as simple as measuring SOC as an indicator of cumulative $CO_2$ sequestration. In this case, the SOC-SIC interactions of interest persisted to 200 cm soil depths, much deeper than is usually sampled in the majority of other studies. Our findings can be applied internationally for other geographic regions with similar climates.

In summary, our data confirms the SOC accumulation found after grassland restorative fencing reported by other studies, as well as the associated loss of SIC reported by Liu and Ping (*Liu et al., 2017*; *Ping & Wang, 2018*). We provide new insight into the SOC accumulation dynamics of the deep loessial soil in a semi-arid climate, and how they continue to evolve over 30 years and to a soil depth of 200 cm.

## CONCLUSIONS

This study compared the vertical distribution of SOC and SIC in a semi-arid loessial soil to a depth of 500 cm at replicated sites under historical continuous grazing (G) or sites where grazing had been fenced off 15 (F15) or 30 (F30) years prior to the study. We confirmed prior findings on SOC accumulation after grazing exclusion, but few, if any, previous studies have sampled below 100 cm soil depths. Our deeper sampling provided new insights and revealed substantial SOC accumulation in the 100–200 cm soil depths 15 to 30 years after fencing. This finding indicates that slow diffusion processes intensify the soil property changes caused by increased litter return after fencing. As previously reported, the SOC accumulation over time is largely offset by SIC loss. The present study found correlations across the dataset between soil properties to clarify that the SOC-SIC antagonism is pH-mediated. The new information from this study may be factored into soil C inventory calculations for loessial soils in similar climates internationally.

## ACKNOWLEDGEMENTS

We thank the all staff members of Guyuan National Nature Reserve for providing field assistance for us to collect soil samples in the Yunwu mountain.

### Funding

This work was supported by Top Discipline Construction Project of Pratacultural Science (NXYLXK2017A01), the Chinese Postdoctoral Science Foundation (2015M580896), and the National Natural Science Foundation of China (31660143, 41501094). The funders had no role in study design, data collection and analysis, decision to publish, or preparation of the manuscript.

### Grant Disclosures

The following grant information was disclosed by the authors:
Pratacultural Science: NXYLXK2017A01.
Chinese Postdoctoral Science Foundation: 2015M580896.
National Natural Science Foundation of China: 31660143, 41501094.

### Competing Interests

The authors declare there are no competing interests.

### Author Contributions

- Yi Zhang performed the experiments, analyzed the data, prepared figures and/or tables, and approved the final draft.
- Yingzhong Xie and Hongbin Ma conceived and designed the experiments, authored or reviewed drafts of the paper, and approved the final draft.
- Le Jing performed the experiments, prepared figures and/or tables, and approved the final draft.
- Cory Matthew conceived and designed the experiments, analyzed the data, prepared figures and/or tables, authored or reviewed drafts of the paper, and approved the final draft.
- Jianping Li conceived and designed the experiments, prepared figures and/or tables, authored or reviewed drafts of the paper, and approved the final draft.

### Data Availability

The raw data are available in a Supplementary File.

### Supplemental Information

Supplemental information for this article can be found online at http://dx.doi.org/10.7717/peerj.8986#supplemental-information.

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
