# Peer review of "Rebuilding soil organic C stocks in degraded grassland by grazing exclusion: a linked decline in soil inorganic C"

_PeerJ, doi:10.7717/peerj.8986_

## Round 0.1 · original submission · Major Revisions

The paper is not well written. The English language should thoroughly revised to ensure that an international audience can clearly understand your text. In addition, the paper lacks research questions, there are problems in the methods and statistical analysis that needs to be addressed. The research impact is not assessed.

Reviewer 1 ·

Basic reporting

This is an interesting studying. But there are some critical problems in this study. Firstly, you haven’t explained the research question. Why did you do this study? And the Introduction only summarized the studies about soil organic carbon and ecological restoration. This is not enough. You should show the gaps, between the previous studies and yours. Highlight your own innovation.
L68-91: Please simplify this paragraph.
L94: The reference style should be revised.
L97: The same problem.
L116-124: Please extend the article review about SIC, which also played an important role in your article.

Experimental design

L148-151 Please adjust those sentences. The “At each of the nine sites,….” was looked too abrupt. You should explain the context.
L185-186 Why use the repeated measures ANOVA? Your time-variable looked like a kind of treatment, not real time-variable. Please check the statistic method. If other studies used a similar method, please cite it.
L187-189 Please simplify the explanation. I suggest you add the F value to the p value’s bracket in the Results.
L198 Which correlation analyses have you used? Pearson or Spearman?
Figure 1 Please detail the study area. Because it only showed the study city. The experimental area was not clean.

Validity of the findings

I think you should ask your colleagues to help you revise the Results and Discussion.
L226 “total soil C” or “total C stock”. It is not clean. Please revised it.
L200-252 The “Soil organic and inorganic carbon” should be reversed with the” Soil bulk density, soil moisture (SWC), and pH”. Because the third part of Results named “Total carbon stock percentages of the soil organic carbon and soil inorganic carbon pools”, which showed more relations with the “Soil organic and inorganic carbon”.
L217 Please present the table or figure one by one. And please put together the results of each table or figure.
L239-240 There is not necessary to repeat the table 4 in p value’s bracket.
L298 It is not necessary showed the name of the table and figure again in Discussion.
L341-343 The words are a bit exaggerated. And the same question showed L47-48. Please revised it.
Table 4 Please show the asterisk which could present the significance. And the software’s name should be discarded.
Figure 2 Why is no error bar disappear? Please explain it.

·

Basic reporting

(1) Very sorry for my own poor English, so no comment here is given on language level of this manuscript.
(2) In Table 4, add * or ** after the values of correlation coefficients
(3) Figure 1, strongly advise to delete the small chart on the top left corner due to the reason well known, and add longitude and latitude in the border around the chart.
(4) Figure 2, the error line should add in the curves of SOC and SIC as Figure 3 did.

Experimental design

(1) The method using a 6 m auger to collect soil bulk density samples should describe more clearly.
(2) For the fenced plots, were the grasses planted by human or appeared and grew naturally? and any fertilizing or ploughing activity happened for the grasses in the fenced plots?
(3) From a statistical point of view, only three fenced years (0a, 15a and 30a) are insufficient to form a real time sequence, thus it is very hard to disclose the real change of soil carbon with the fenced year, the convincingness of the obtained results is insufficient. It is strongly advised to add 1-2 other fenced years to the time sequence!

Validity of the findings

no comment

Additional comments

The other following problems need to consider,
1. Ningxia in China is “autonomous region”, not “province”.
2. Why defined the depth as 500 cm, can the grass roots appeared in this study grow down to 500 cm or can the 425 mm precipitation can move down to 500 cm? If so, why seems no change happened to carbon below 200 cm as the study found.
3. mountain gray cinnamon and black loess, in my opinion, not as Spodosols and Inceptisols in USDA Soil Taxonomy, more likely as Alfisols or Inceptisols and Mollisols.
4. What is real soil types in the three kinds of studied plots? Is there any difference in the grass composition in the studied three kinds of studied plots? what is the method to get the grass coverages of the studied plots?
5. Table 1, there are obvious differences in the slopes even in the plots with the same fenced year, is there information available on the influences of different slopes on soil carbon?
6. The unit currently used for SOC and SIC is g kg-1, not %.

---

## Round 0.2 · Minor Revisions

The paper has been revised, although there are still minor issues.

Especially, the English needs a revision.

Some claims are not substantiated, please remove them:

- "This research provides new approaches to the calculation of greenhouse gas inventories" should be removed as this paper does not deal with GHG
- "An increase in SOC correlated with increased soil water holding
capacity" NOT TRUE, no water holding capacity is measured. The authors only measure soil water content at one time, which does not reflect capacity and also dynamics.

Reviewer 1 ·

Basic reporting

The article has been well revised. There were only some small errors.

Experimental design

L180 and 185: in my opinion, it may be better to replace the "repeated measures“ by “mixed effect model”.

L186: the “x” should be replaced by “×” or “and”.

Validity of the findings

Figure 1B is it “C” or “c”

Please revise the Figure 2B and Figure 3B. The letters were too indistinct and ugly.

Figure 5
The figure’s orders should use “A)” which is consistent with previous figures.

---

## Round 0.3 · accepted · Accept

The paper has been revised accordingly and ready for publication. Congratulations.